# Integrated bioinformatics and statistical approach to identify the common molecular mechanisms of obesity that are linked to the development of two psychiatric disorders: Schizophrenia and major depressive disorder

Md Khairul Islam[1], Md Rakibul Islam[1], Md Habibur Rahman[2], Md Zahidul Islam[1], Md Mehedi Hasan ●[3], Md Mainul Islam Mamun[4], Mohammad Ali Moni ●[5]*

1 Dept. of Information Communication Technology, Islamic University, Kushtia, Bangladesh, 2 Dept. of Computer Science Engineering, Islamic University, Kushtia, Bangladesh, 3 Department of Statistics, University of Rajshahi, Rajshahi, Bangladesh, 4 Department of Applied Physics and Electronic Engineering, University of Rajshahi, Rajshahi, Bangladesh, 5 Dept. of Computer Science and Engineering, Pabna University of Science and Technology, Pabna, Bangladesh

* m.moni@uq.edu.au

**Data Availability Statement:** We have collected the patient data from National Center for

## Abstract

Obesity is a chronic multifactorial disease characterized by the accumulation of body fat and serves as a gateway to a number of metabolic-related diseases. Epidemiologic data indicate that Obesity is acting as a risk factor for neuro-psychiatric disorders such as schizophrenia, major depression disorder and vice versa. However, how obesity may biologically interact with neurodevelopmental or neurological psychiatric conditions influenced by hereditary, environmental, and other factors is entirely unknown. To address this issue, we have developed a pipeline that integrates bioinformatics and statistical approaches such as transcriptomic analysis to identify differentially expressed genes (DEGs) and molecular mechanisms in patients with psychiatric disorders that are also common in obese patients. Biomarker genes expressed in schizophrenia, major depression, and obesity have been used to demonstrate such relationships depending on the previous research studies. The highly expressed genes identify commonly altered signalling pathways, gene ontology pathways, and gene-disease associations across disorders. The proposed method identified 163 significant genes and 134 significant pathways shared between obesity and schizophrenia. Similarly, there are 247 significant genes and 65 significant pathways that are shared by obesity and major depressive disorder. These genes and pathways increase the likelihood that psychiatric disorders and obesity are pathogenic. Thus, this study may help in the development of a restorative approach that will ameliorate the bidirectional relation between obesity and psychiatric disorder. Finally, we also validated our findings using genome-wide association study (GWAS) and whole-genome sequence (WGS) data from SCZ, MDD, and OBE. We confirmed the likely involvement of four significant genes both in transcriptomic and GWAS/WGS data. Moreover, we have performed co-expression cluster analysis of the

Biotechnology Information (NCBI) GEO datasets with the accession number: GSE95243, GSE92874 and GSE125664.

**Funding:** The author(s) received no specific funding for this work.

transcriptomic data and compared it with the results of transcriptomic differential expression analysis and GWAS/WGS.

## Introduction

Obesity is a chronic condition with many associated health complications arising due to the large-scale accumulation of body fat. It is an underlying health issue that increases the risk of acquiring other diseases and health problems, including cardiovascular disease, osteoarthritis, type II diabetes, stroke, high blood pressure, sleep apnea, hypertension, and certain types of cancer [1]. Thus, the dangers of obesity are being exacerbated by its linked disorders. Unfortunately, Obesity has increased in prevalence in recent times. As of 2016, almost 1.9 billion adults (age > 18) on earth were overweight among which 0.65 billion were affected by obesity [2], numbers that have nearly tripled since 1975 [3]. It is also predicted that by 2030 almost half of the population will be obese [4]. Obesity is also becoming common in children aged 5 to 19 years. In 2016, 330 million children were affected by obesity or were overweight, which has been increasing as a linear trend [5]. Biological, psychological, and environmental factors contribute to the development of obesity as well as individual dietary overconsumption [6, 7]. Excessive appetite is a genetic trait regulated by genes that affect energy consumption and utilisation [8, 9]. Obesity itself has been recognized as a disease by The Obesity Society (TOS) and American Medical Association(AMA) in 2008 and 2013, respectively [10–12], because it is associated with multiple disorders, displays distinct morbidity, raises mortality, and declines health-related quality of life (HRQOL). Thus, it is unavoidable that obesity and its associated diseases will be the most frequently occurring diseases all around the world in recent future [13]. Therefore, it is high time to find out the underlying biological mechanism between obesity and psychiatric disorder.

Symptoms or associated diseases commonly associated with obese patients at an elevated rate compared to non-obese people are psychiatric disorders (including depression and anxiety) [14], sleep apnea, breathing problems, bipolar disorder [15], cardiovascular diseases [16], and gastrointestinal diseases. [17]. Obesity has been linked to psychiatric diseases, particularly major depressive disorder, according to epidemiological and meta-analytic studies [18, 19]. Maternal obesity, on the other hand, has a long-term effect on offspring's neurodevelopment. Evidence from the last few decades suggests that a mother's poor diet and obesity are linked to a variety of psychiatric disorders in her children, such as depression, schizophrenia, hypertension, and anxiety [20–22].

Schizophrenia is a mental illness that impairs many aspects of cognition, reasoning, and behaviuor [23, 24]. Symptoms such as hallucinations, disordered thoughts or behaviours, delusions, and difficulty in focus and concentration, are frequently seen in patients suffering from this condition [25]. The symptoms vary, but usually appear in early adulthood and worsen over time [26]. Additionally, neuroimaging researchers have found evidence that schizophrenia patients have significant features alterations in the central nervous system (CNS); and CNS dysfunction consistent with being a primary brain disorder [27–30]. Likewise, obesity is responsible for metabolic dysfunction and inflammation which lead to damage of CNS system [31, 32]. Recent evidence showed that schizophrenia patients have higher risk of weight gain due to antipsychotic medication or chronic situation [33, 34]. As a result, understanding the similar connections between obesity and schizophrenia can aid in the development of therapeutics for the association [35].

Major depressive disorder (MDD), commonly referred to as depression, is also a mental condition that lasting for 14 days or more [36]. Delusions or hallucinations may occur in MDD patients, just as they do in schizophrenia patients. There are, however, some patients who only have short-term MDD symptoms, while others have symptoms that are prolonged and are more severe [25, 37]. Dysregulation in neurotransmitter levels may play a role in depression since alterations in neurotransmitter levels can affect mood stability [38]. Similarly, neurotransmitter level controls the weight gain, so dysfunction in neurotransmitter cause overweight in individuals [39]. Obesity and depression has bidirectional relationship not only due to enviornmental issues but also others factors such as dysfunction in neurotransmitter, oxidative stress, low inflammation and neuroendocrine regulators [35, 40, 41]

In one study of 841 individuals of Chinese ethnicity seeking obesity treatment, almost 42% were diagnosed with at least one psychiatric disorder [42], and such findings are not unusual. While clinical, epidemiology, morbidity, and other data indicate that obese patients are at risk of being affected by a psychiatric disorder, it is unclear whether gene pathways influencing obesity are also involved with the psychiatric disorders seen in these patients. There are many environmental variables that contribute to obesity's development. Obese individuals often live in 'obesogenic' environments with ready access to calorie-dense high sugar and high-fat foods commonly seen in industrialized countries [43].

To define the genetic factors of obesity and its association with other disease, most studies examine neurotransmitter level, inflammation and endocrine systems etc. Obesity is associated with dysregulation of endocrine factors including a range of pituitary hormones, thyroid hormone, leptin, resistin [44] and key inflammatory cytokines interleukin-6 (IL-6), and tumour necrosis factor-α (TNF-α) [45]. In obese infants, the pituitary gland modulate the function of the hypothalamic-pituitary-adrenal (HPA) by cortisol clearance [44]. Monogenic disorder like Bardet–Biedl syndrome (BBS), consist of almost 20 genes, showing obesity symptom due to protein integrity or mutations in the genes. Single gene like Melanocortin-4 receptor (MC4R) mutations blamed for obesity in many research [8, 46]. Genes associated with obesity control several metabolic pathways (e.g., leptin and melanocortin) that are known to regulate complex interactions with neural circuits of the central nervous systems (CNS) and affect neurotransmitter and neuropeptide levels [8, 43]. Moreover, 222 studies of obesity gene map identified 71 candidate genes for obesity association [47].

Endocrine system dysfunction and neurological psychiatric disorders have multifaceted unknown relation which is yet needed to be unveiled [48]. Due to the effect of hyperinsulinism, Somatomedin (Insulin-like growth factor-1) growth in large count caused MDD in men [49]. The increasing pituitary volume of the HPA axis is correlated with the development of abnormalities/psychosis in susceptible individuals. Similarly, it is induced by triggering the hormonal stress response provoked by hormonal changes in obese patients [50].

Little is known about the biological pathways that are associated with both obesity and psychiatric disorders. However, previous researches identified pathways associated with them such as the kynurenine, neurotransmitter, immune inflammation, neuroprogression, and HPA pathways [51]. Several neurotransmitter pathways are dysregulated in both obesity and MDD [52] such as the kynurenine pathway [53], and the oxidative stress pathways. Indeed, increased oxidative stress related to NADPH (Nicotinamide adenine dinucleotide phosphate) oxidase stimulation results in creating increased levels of adipocytokines [53]. Oxidative stress and redox imbalances are also known to affect mental conditions such as anxiety, schizophrenia, MDD, and bipolar disorder [54, 55].

In summary, there is compelling evidence of pathological associations between psychiatric disorders and obesity that are not well understood. The goal of this research is to find relevant connections in order to know more about the disorders' functions. We thus developed a

pipeline that integrates bioinformatics and statistical approaches to analyze transcriptomic, whole-genome sequencing (WGS) and genome-wide association study (GWAS) data of the aforementioned diseases to determine pathways they have in common. We have collected the transcriptomic data called Expression profiling by high throughput sequencing (RNA-seq) from GEO database (https://www.ncbi.nlm.nih.gov/gds). All of the transcriptomic datasets are built using induced pluripotent stem cell (iPSC) technology. Unfortunately, in vivo data related to brain/hypothalamic tissue is hard to find. In addition, significant gene extracted from GWAS/WGS are not enough to prove the risk of a disease [56]. Therefore, the identification of important genes from living organs/cells is therefore aided by patient-specific iPSC data. Moreover, because the human brain functions as a single mechanism, iPSC technology contributes in elucidating critical etiologies of psychiatric disorders [56, 57]. On the other hand, hypothalamus consists of neurons that are responsible for appetite in human and reliable for dysfunction in endocrine system. As a result, obesity data gathered from hypothalamic tissue could provide more information about the molecular mechanisms that lead to obesity. However, due to the lack of hypothalamic neurons-related data from living organs, iPSC technology was applied [58, 59]. Additionally, we gathered GWAS and WGS data from the following well-known databases: GWAS catalog [60], PheGenI [61], dbGaP [61], UK-Biobank [62] and Clinver (from NCBI). We thus identified genes significantly associated with obesity and then compared them with schizophrenia and MDD. By performing multiple functional analyses and data mining on these genes we identified 410 genes and 199 pathways altered in obesity that are also dysregulated in schizophrenia and/or MDD. This study thus identified pathways that may interact in obese patients with schizophrenia or MDD, suggesting possible future approaches to therapeutic intervention.

$$P_Y = \underbrace{H(Y_n) - H(Y_n|\mathbf{V}_n^Y)}_{S_Y} + \underbrace{H(Y_n|\mathbf{V}_n^Y) - H(Y_n|\mathbf{V}_n^{X,Y})}_{T_{X \to Y}}, \tag{1}$$

## Materials and methods

### Data preprocessing and biomarker genes identification

Obesity patient iPS cell RNA-seq transcriptomic datasets were examined in this study. The dataset was generated from induced pluripotent stem cells of hypothalamic/motor neurons of obese patients and completed by the Board of Governors Regenerative Medicine Institute in Los Angeles, California [59]. The dataset's GEO accession number is GSE95243, and all the curated datasets are in the public domain, namely the NCBI database (https://www.ncbi.nlm.nih.gov/). The study includes 7 cell samples derived from hypothalamic neurons from healthy individuals and 2 cell samples derived from motor neurons. Case study 5 samples were collected from obese patients and 5 samples were obtained post-mortem, both derived from hypothalamic neurons. RNA-seq transcriptomic data extracted from the samples (using QIA-GEN RNeasy mini kits), sequenced, and compared between control vs case samples to obtain insights into the transcriptional dysfunction in the sampled cells from diseased individuals.

Aside from obesity-related data, we have analyzed RNA-Seq transcriptomic data from schizophrenic individuals (and normal controls) from NCBI with the GEO accession number **GSE92874** [63]; researchers of SUNY Pathology and Anatomical Sciences Department produced the data. The study included 4 healthy patient samples and 4 schizophrenia patient samples; the samples were generated using iPS cells. Another study similarly employed RNAseq data using iPS cells from MDD and control individuals; the data were collected from NCBI with the GEO accession number GSE125664 [64]. This study was conducted by Max Planck

Institute of Immunobiology and Epigenetics, Freiburg im Breisgau, Germany. Six major depressive disorder patients and three healthy patient samples were collected from iPSC-derived neurons.

The datasets were processed to identify genes that are differentially expressed in individuals affected by the conditions and their respective unaffected controls. RNA-seq transcriptomic data was processed by the DESeq2 package using a negative binomial distribution assumption, and the differential expression was determined from RNA data processed by the limma package. We then performed quantile normalization to eliminate platform technology-related variation and data noise [30]. The differentially expressed genes were then filtered using two conditions to identify the most important genes. The P-value thresholds of less than 0.05 were the first one, and the second was an absolute log2 fold change greater than 1 (— log2fc — $\leq$ 1).

Diseasome networks were constructed to identify possible disease-genes interactions. And it identify the connection between disease and gene using the topological-based statistical method. Such network was developed using Cytoscape V 3.8.2 [65] where nodes define either a disease or a gene. For the diseasome network, a collection of diseases (D) and corresponding datasets (G) are used, with each gene g *in* G linked with a specific diseased *in* D. Suppose, we have two sets of genes $G_n$ and $G_m$ both from either up and down-regulated genes that are linked with the disease $D_n$ and $D_m$, respectively. Thereby, the total number of common significant genes ($n_{nm}$) linked with both $D_n$ and $D_m$ was identified by:

$$n_{nm} = N(G_n \cap G_m) \tag{2}$$

Where, $n_{nm}$ denote the total number of shared genes between obesity and psychiatric disorder (SCZ and MDD). The shared genes are either up or down regulated. Moreover, We also identified the shared up regulated genes and shared down regulated genes between obesity and a psychiatric disorder.

## Specific genetic variations and data mining approaches to identify genetic markers shared by obesity, schizophrenia, and MDD in GWAS and WGS studies

Genome-wide association study (GWAS) involves studying a set of genetic variations linked to the occurrence of a particular disease or condition. In this process, we extracted and analyzed the genomic sequences for many different individuals to identify the DNA polymorphisms occurring in the genome of individuals. Numerous meta-analyses of genome-wide association studies (GWAS) data were collected from obese and control patients. Researchers had first collected DNA from individuals affected by a specific condition (and unaffected controls) and then analyzed genetic variants using single nucleotide polymorphism (SNP) arrays to identify the variant SNPs associated with the development of a disease [66]. Using this GWAS data, we identified 283 genes that associate significantly with obesity when the p-value is less than 1.0E-5. We also looked at the GWAS/WGS data to see whether any genes were linked to the development of psychiatric disorders like schizophrenia and major depressive disorder. We examined several genetic research catalogs, such as the GWAS catalog [60], PheGenI [61], dbGaP [61], UK-Biobank [62] and Clinver from NCBI (for WGS), as part of our overall genomic research strategy. We then compared significant genes identified from GWAS and WGS database with significant genes from the obesity transcriptome dataset.

## Signalling pathways and gene ontology

We conducted functional enrichment analysis for gene ontology, a statistical procedure for determining genes that may influence interactions between obesity and psychiatric disorders.

This method also identifies the cell pathways that influence a particular shared cell function. We accomplished the task by utilizing Enrichr (https://maayanlab.cloud/Enrichr/) tools [67]. We explored the BioCarta-2016 [68], KEGG-2019-Human [69], Reactome-2016 [70], and WikiPathways-2019-Human [71] databases for enriched signalling pathways and performed ontological analysis using the GO Biological Process (2018) database. In this research, a P-value (using Benjamini Hochberg's (BH) procedure [72]) $\leq 0.05$ was used to identify the most significant signalling pathways and gene ontologies. Then we have combined the significant pathways from all the databases to determine the top 25 pathways. We also represented the top 25 significant GOs.

## Obesity and psychiatric disorders—Gene expression correlations

We then evaluated the genetic contribution of obesity-related significant genes connected with psychiatric diseases. The likelihood that a random gene is responsible for a disease-disease association is described by the hypergeometric distribution. Assume that N is a collection of all genes in the genome, and M is the number of significantly expressed genes associated with obesity. The number of significant genes (s) is extracted from psychiatric disorders, and it is discovered that k genes are shared by obesity and psychiatric disorders. Then, the expression denotes the hypergeometric probability is:

$$h(k; N, s, M) = \frac{\left[\binom{M}{k}\binom{N-m}{s-k}\right]}{\binom{N}{s}} \tag{3}$$

The mean and variance of the distribution are $n * M/N$ and $s * M * (N-M) * (N-s)/[N^2 * (N-1)]$, respectively.

The hypergeometric distribution identifies the probabilistic value, expressing disease-disease associations at the genetic level. We also applied the same hypergeometric distribution to GO/pathways, where N indicates the total number of GO/ pathways, M indicates the total number of GO/pathways related to obesity, s denotes the total number of GO/pathways related to a psychiatric disorder, and n implies the total number of common GO/pathways associated with both obesity and psychiatric disorders. Additionally, we have calculated the Jaccard Index (i.e., the Jaccard similarity coefficient) to reflect the relationship of disease at the pathway level [73]. If A represents the total number of GO-pathways linked to obesity, B represents the total number of GO-pathways linked to psychiatric disorders, and C represents the number of shared GO-pathways between obesity and a psychiatric disorder, then the Jaccard Index is defined as follows:

$$J = \frac{C}{A + B - C} \tag{4}$$

## Cluster and co-expression analysis of the transcriptomic profile

Firstly, we have considered all the significant DEGs and SNPs associated with obesity from transcriptomic, GWAS, and WGS data. Similarly, we have included biomarker genes and SNPs of schizophrenia and major depressive disorder (MDD); and then we have compared them to those with obesity in order to find genes of interest; and perform co-expression and cluster analysis of transcriptomic profiels. We previously used semantic similarity-based

analysis to identify comorbid conditions between systemic sclerosis and cancers [74]. In differential expression analysis, both healthy and disease samples included for the analysis that produce comparative expression level of the data. In contrast, We have performed the co-expression on the transcriptomic raw data of obesity and psychiatric diseases which are sequence count data of a particular disease state only [75]. Pearson correlation has been applied to all the identified genes of interest for the diseases. The transcriptome data samples serve as characteristics/features of the selected genes that contain count values for relevant genes. Finally, we constructed a hierarchical cluster of the co-expressed data to show the most significant genes of interest with the highly expressed transcripts.

## Results

The datasets we employed for our analysis were collected from publicly available resources including transcriptomic analysis of RNA-Seq data as well as GWAS and WGS data to analyze genetic variants. Fig 1 shows the sequence of our methodological approach.

### Expressed biomarker genes analysis of obesity and psychiatric disorders using data from iPSC-derived neuronal cells

In our experiments, we have used RNA-seq data to assess iPSC gene expression data and the data was collected from both patients and healthy people. These iPSC transcriptome profile identified deferentially expressed genes in individuals with these conditions relative to control individuals. Fig 1A shows the gene expression profiling from three types of patients: obese patients, patients with MDD, and with schizophrenia. And Fig 2A–2D illustrates the correlated genes that are differentially expressed in obesity and psychiatric disorders (MDD and schizophrenia).

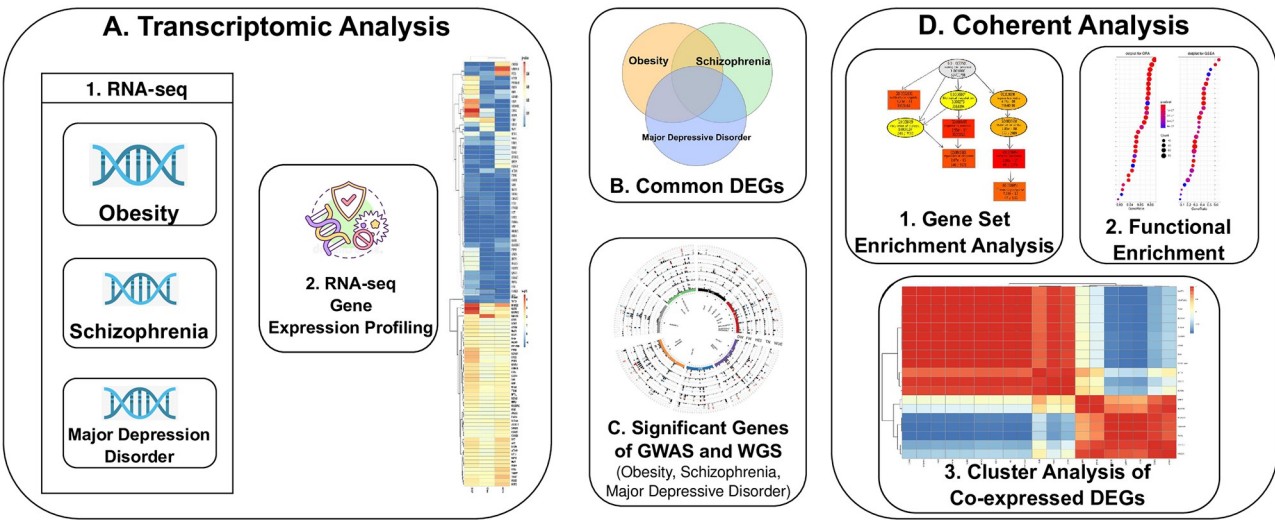

**Fig 1.** A. Transcriptomic analysis of obesity, MDD, and schizophrenia RNA-seq (high throughput sequencing) datasets. 1. Collection of RNA-seq (high throughput sequencing) data from individuals with obesity, schizophrenia, and MDD respectively. 2. Gene expression profiling of iPSCs for the selected RNA-seq transcriptomic data. 3. Visualisation of deferentially expressed genes in tissues from individual patients with the disease. B. Venn diagram of the significant DEGs identifying the common genes between obesity and psychiatric disorder datasets. C. Filtered genes from different whole-genome sequences (WGS) and GWAS databases regarding the selected diseases from this study. This is followed by the determination of the significantly different genes using criteria based on p-value (< 1.0E-5). D. Different approaches to testing the hypothesis. 1-2. Gene-Set Enrichment and Functional Enrichment analysis to verify that obese patients have the possibility of developing a psychiatric disorder. 3. Cluster analysis of the shared co-expressed DE-Genes between obesity and psychiatric disorders, while count data collected from the selected transcriptomic samples of our studies.

We applied two filtering conditions to separate significant genes which are immensely def-erentially expressed in affected patients relative to control patients, namely log2-fold change > 1 (absolute log2-fold change) and p-value < 0.05 for the differences. Then, the filter genes are termed as 'significant genes', and using these, we compared the up and down-regu-lated genes seen in the obese patient with those of schizophrenia and MDD patients. 51 genes were found to be shared by all three diseases, 49 of which were up-regulated, meaning that their expression changed in the same way in all three disorders, and two of which were down-regulated. Of the 1517 significant genes expressed in obesity 357 up or/and down-regulated genes are found in common with either schizophrenia or MDD as shown in Fig 1. Obesity patient's transcriptomic profiles shared 162 (114 up-reguated and 49 down-regulated) and 246 (186 up-regulated and 61 down-regulated) dysregulated genes with those with schizophrenia and major depressive disorder, respectively. Later in these experiments, we analyzed the shared genes profiles to find the common molecular mechanism that exists both in obese patients and those suffering from psychiatric disorders as shown in Figs 5 and 6.

In Fig 2, A heatmap (using p-value and log2-fold change) and bubble plot is introduced using shared dysregulated genes among all the diseases in our study. This visual representation and ranking of shared genes shows the striking nature of the transcriptional effect triggered in obese patients as well as in mental patients. Such as log fold change describe the alterations and biological relevance in patient due to the development of a disease [76].

## GWAS study of obesity patients that have common biomarker genes with schizophrenia and MDD patients

We have explored obesity patient's meta-analysis GWAS and WGS data from various sources. In this study, we looked at 1366 raw genes and found 1130 significant genes that had a p-value < 1.0E-5 (Fig 3). We have considered our GWAS and WGS from GWAS catalog [60], PheGenI [61], dbGaP [61], UK-Biobank [62] and Clinver from NCBI (WGS), etc. We also identified genes from GWAS and WGS databases for our selected psychiatric disorders. Pri-marily, we have examined 1961 and 446 raw genes, respectively, in the context of schizophre-nia and depression. The condition, p-value is less than 1.0E-5, also applied for schizophrenia and major depressive disorder, where we found 1836 and 406 significant genes, respectively. Then, genes associated with obesity and psychiatric disorder (as identified by GWAS and WGS) were analyzed to find shared genes among them. To be exact, 1130 significant genes associated with obesity where 126 and 43 genes are commonly shared and associated with schizophrenia and MDD respectively (as shown in Fig 3). Only 13 genes are common to all the selected diseases: AL136114, WWOX, PCDH9, TCF4, ZNF536, RBFOX1, MIR924HG, CACNB2, SEMA3A, TENM4, DPP10, PTPRD, and MPP6. Fig 3 illustrates the relationship between diseases by utilizing highly expressed common genes from our sampled data (GWAS and WGS). Finally, we have compared shared significant genes identified through GWAS/WGS for the two pairs of relationship (obe and scz/ obe and mdd) to transcriptomic profiles for the same pairs. As a result, three genes are commonly found between obesity and schizo-phrenia both from transcriptomic expression profiles and GWAS/WGS analysis, and one gene commonly identified between obesity and depression.

## Diseasome network between obesity and psychiatric disorders

Comparative analysis was performed to identify shared differentially expressed genes between the psychiatric disorders and obesity datasets. In Fig 2A, There were 49 up-regulated genes and two down-regulated genes that were found to be shared between psychiatric disorders and obesity profiles, all of which are significantly expressed in both conditions. In addition, the

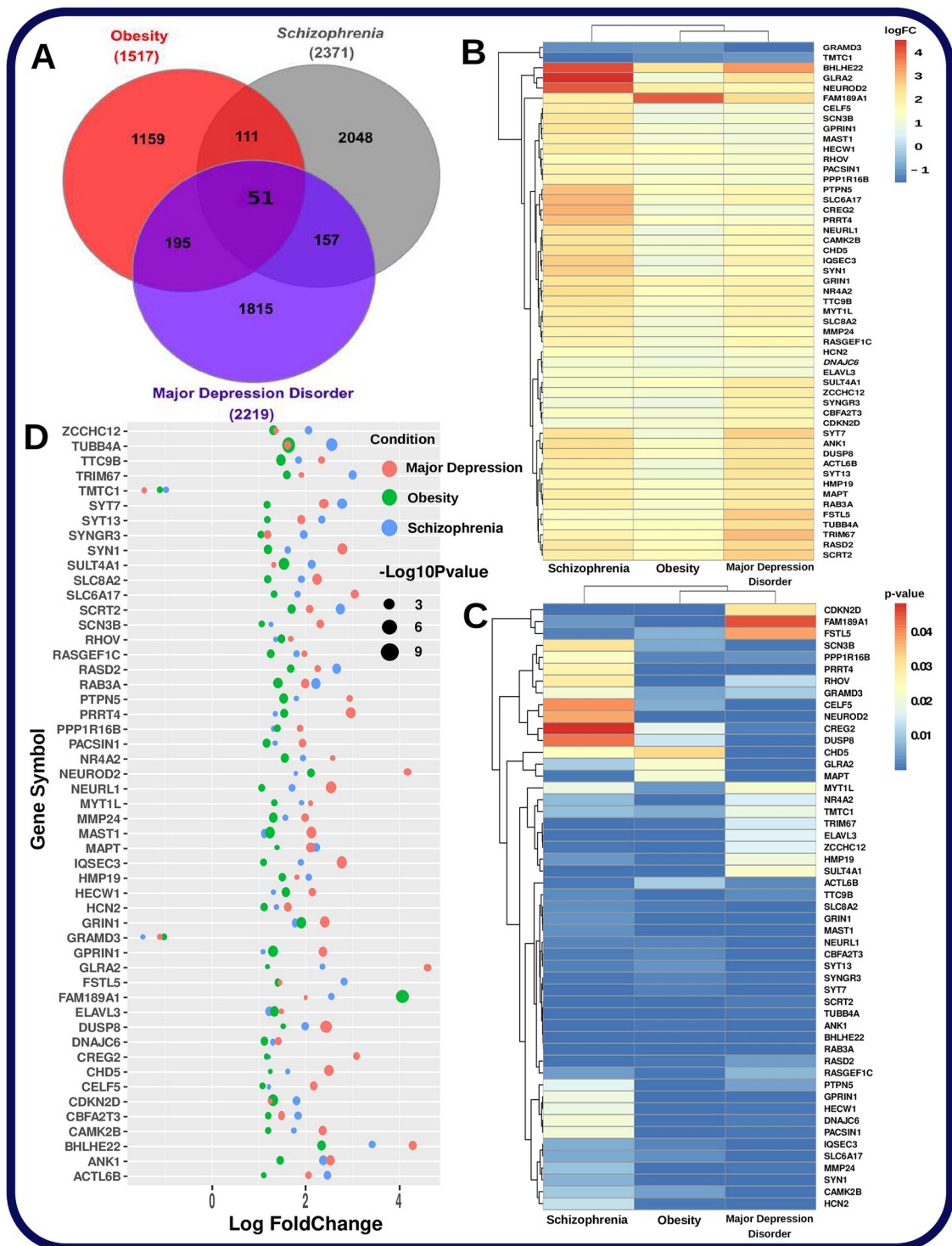

**Fig 2.** A. Venn diagram shows the significant common biomarker genes of obesity along with schizophrenia and MDD from transcriptomic data iPSCs derived from affected patients and controls. B. Heatmap illustrated using the log2-fold changes of the shared significant genes between obesity and psychiatric disorders. C. Heatmap constructed using the p-values of the common significant genes between obesity and the two psychiatric disorders. D. Log-fold changes and p-value combined to generate a bubble plot for the common significant genes between obesity and psychiatric disorders.

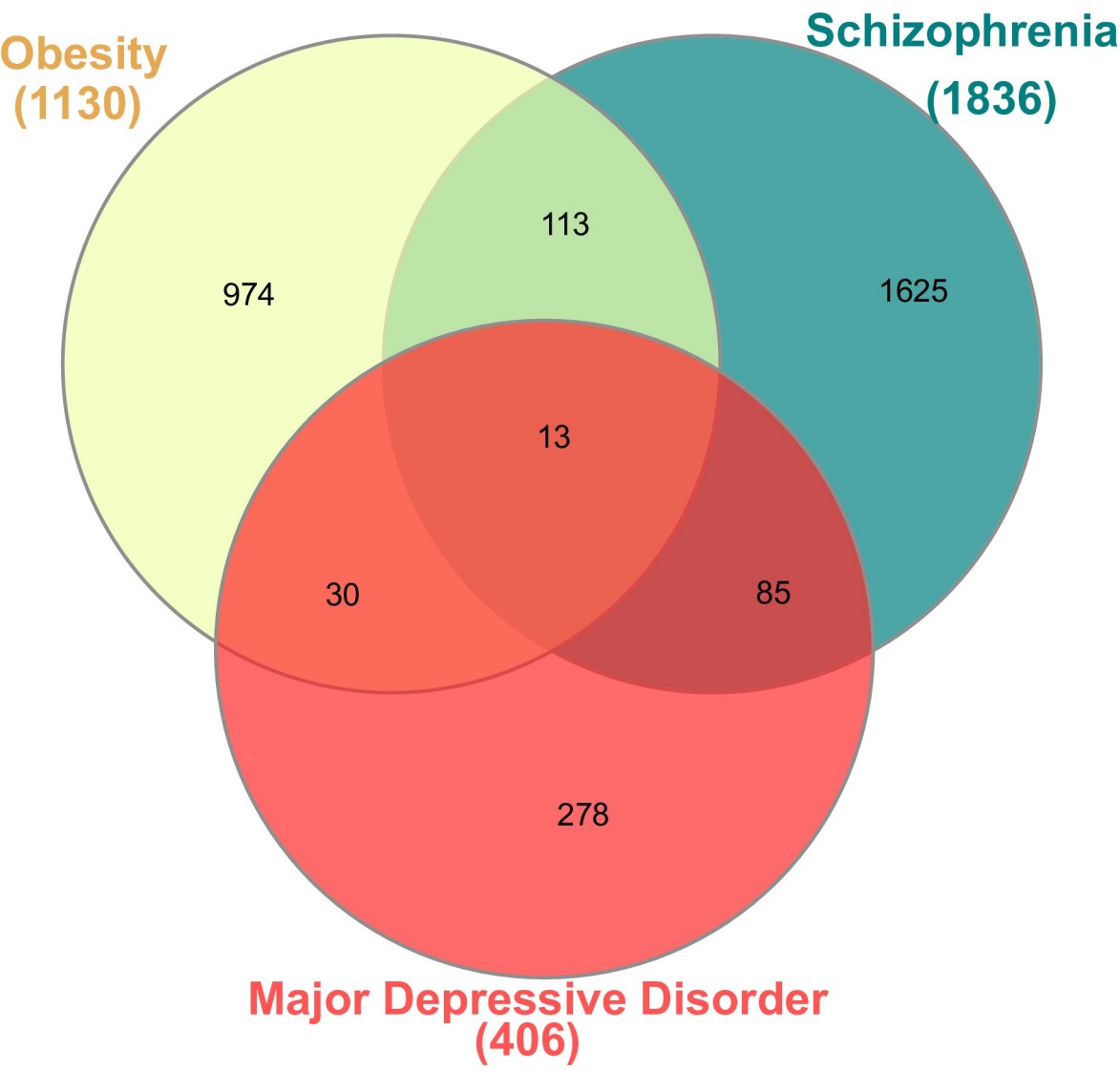

**Fig 3. The Venn diagram identifies shared significant genes between obesity and psychiatric disorders (schizophrenia and major depressive disorder) based on WGS and GWAS data from several sources: GWAS catalog, PheGenI, dbGaP, UK-Biobank and Clinver (from NCBI) etc.**

obesity dataset shared 34 up-regulated and 10 down-regulated genes with schizophrenia. Similarly, obesity and depression shared 68 up-regulated and 11 down-regulated genes between them. We then created a diseasome network shown in Fig 4 that shows and mention the common up and down-regulated genes among all the three diseases. Apart from the 51 genes that are found in all diseases, it indicates the number of shared up and down regulated genes between obesity and a particular psychiatric disorder. The most significant common up-regulated genes are: ACTL6B, ANK1, BHLHE22, CAMK2B, CBFA2T3, CDKN2D, CELF5, CHD5, CREG2, DNAJC6, DUSP8, ELAVL3, FAM189A1, FSTL5, GLRA2, GPRIN1, GRIN1, HCN2, HECW1, HMP19, IQSEC3, MAPT, MAST1, MMP24, MYT1L, NEURL1, NEUROD2, NR4A2, PACSIN1, PPP1R16B, PRRT4, PTPN5, RAB3A, RASD2, RASGEF1C, RHOV, SCN3B, SCRT2, SLC6A17, SLC8A2, SULT4A1, SYN1, SYNGR3, SYT13, SYT7, TRIM67,

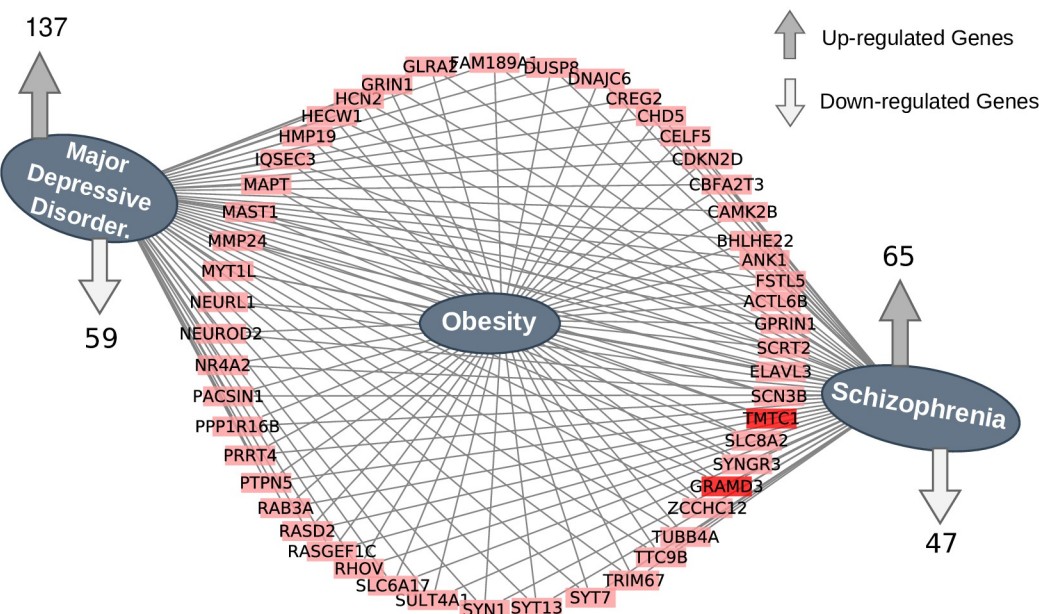

**Fig 4. The figure illustrates the up- and down-regulated genes that are shared by obesity and psychiatric illnesses.** Genes are represented by a rectangle-shaped pink/red color, while diseases are represented by an ellipse-shaped indigo color. Genes with a pink color are up-regulated, whereas those with a red color are down-regulated.

TTC9B, TUBB4A and ZCCHC12. The most significant common down-regulated genes are GRAMD3 and TMTC1.

## Gene-ontology and cell-signalling pathways analyses between obesity and psychiatric disorders

To examine the biomarker genes of obesity and psychiatric disorders, we used transcriptomic data and identified the common genes between obesity and schizophrenia/MDD. Then, we have performed gene set enrichment analysis using the shared genes through cell signalling pathway and gene ontology to get biological meaning for the diseases. To accomplish that, the well-established databases of BioCarta, KEGG, WikiPathways, Reactome, and Gene Ontology databases were used. Transcirptomic Obesity data shared 162 and 246 genes with schizophrenia and depression respectively. Thus, we have used transcriptomic shared significant genes to find the corresponding pathways and GO term. We have found 385 shared signalling pathways between obesity and schizophrenia; and 530 shared pathways between obesity and depression. Moreover, obesity shared 452 and 843 GO terminologies with scz and mdd, respectively. Then, P-value $\leq 0.05$ conditioned applied to get most significant GO term of biological process and signaling pathways for schizophrenia and MDD that are also found in obesity patient datasets. Following the condition obesity have 48 and 151 shared significant pathways with scz and mdd, respectively; as well as 89 and 209 shared significant GO terms sequentially. However, in Figs 5 and 6, we only represent the top 25 signalling pathways and GO terms considering the p-value condition.

GWAS and WGS data also utilised to identify the GO term of biological process and signalling pathways. Obesity have 126 and 43 shared significant genes with scz and mdd, respectively. Obesity shared 535 signalling pathways and 943 GO terms with schizophrenia among which 50 significant singnalling pathways and 168 significant GO tems. P-value ($< 0.05$) is

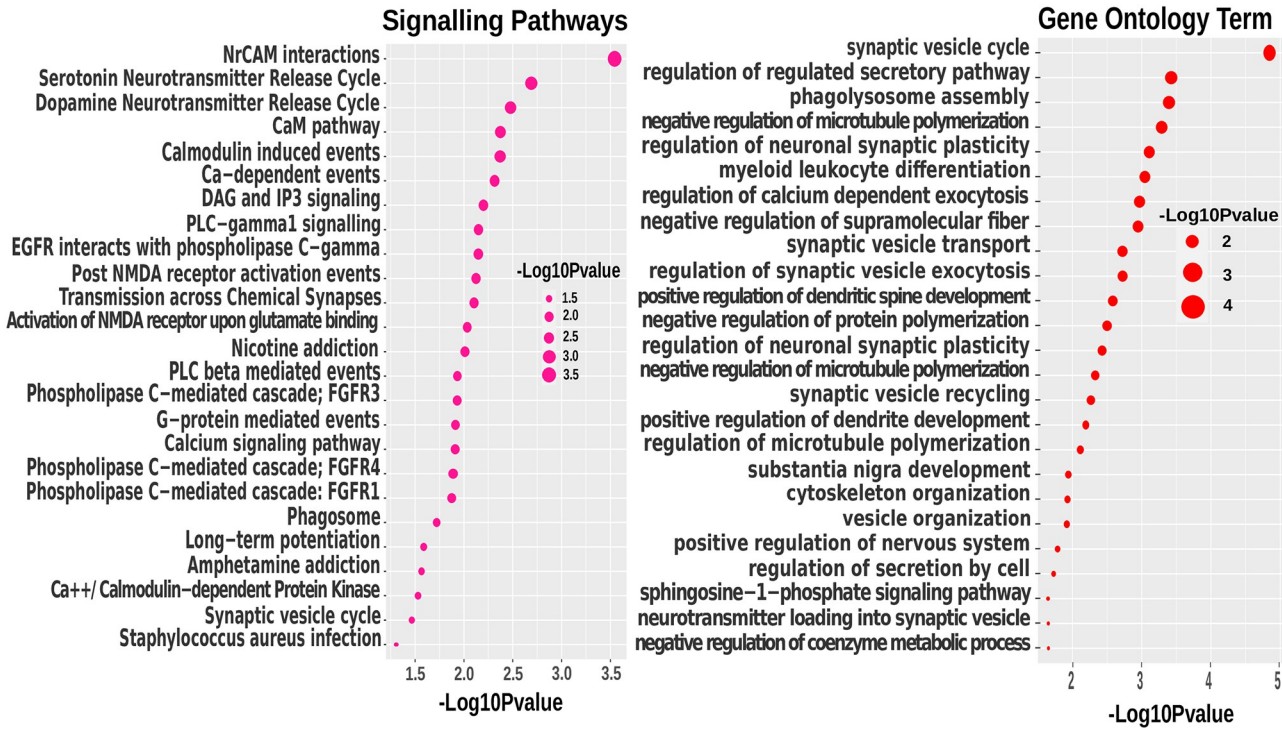

**Fig 5. Gene set enrichment analysis of shared genes between obesity and schizophrenia through signalling pathways and GO to identify pathogenic mechanisms that may link schizophrenia and obesity.**

used to define the significant pathways and GO. Likewise, obesity shared 243 pathways and 301 GO shared with depression. The condition reduce the number to 42 and 92 significantly shared signaling pathways and GO terms, respectively. At last we have compared the shared pathways and GO terms related results of transcriptomic analysis with that of GWAS/WGS shared results.

## Obesity and psychiatric disorders genetic correlations

The hypergeometric distribution was applied on transcriptomic and GWAS/WGS data to examine the common gene's genetic significance in obesity and schizophrenia as shown in Table 1. We have identified that 162 genes are common both in obesity and schizophrenia, and 246 genes are common in both obesity and Major depressive disorder. The hypergeometric distribution test confirms the significance of the identified genes for schizophrenia (p-value = $1.01 \times 10^{-26}$) and MDD (p-value = $1.50 \times 10^{-70}$) are significantly associated with obesity at the gene level. Furthermore, GWAS and WGS data indicate that obesity shared 126 genes with schizophrenia and 43 genes with major depressive disorder. It also shows that both schizophrenia (p-value = $1.1 \times 10^{-568}$) and MDD (p-value = $3.2 \times 10^{-244}$) significantly associated with obesity. The above analysis depicts that both schizophrenia and major depressive disorder are significantly associated with obesity in both transcriptomic and DNA levels.

The association of gene expression pathway patterns seen in obesity and the two psychiatric disorders at the pathway level can be represented by a hypergeometric distribution and Jaccard Index for the transcriptomic and GWAS/WGS data as shown in Table 2. In this study, we didn't considered shared genes between obesity and psychiatric disorders, rather we have analysed pathways determined by significant genes of either obesity or psychiatric disorders. We

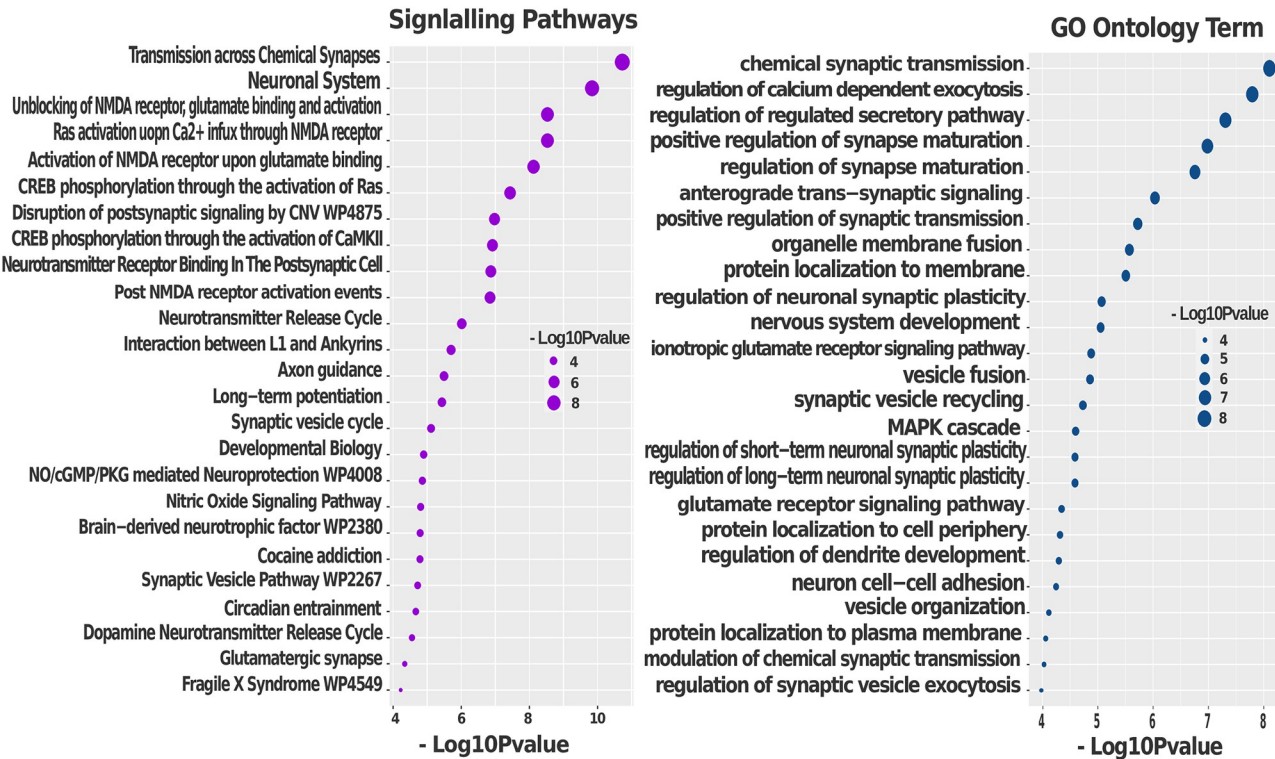

**Fig 6. Gene set enrichment analysis of shared genes between obesity and MDD through signalling pathways and GO to identify mechanisms linking MDD and obesity.**

identified 134 significant pathways shared by both obesity and schizophrenia, and 65 significant pathways are shared by both obesity and MDD. Thus, the hypergeometric test reveals that both schizophrenia (p-value = 1.45 x $10^{-88}$ and Jaccard index = 0.30) and MDD (p-value = 1.12 x $10^{-19}$ and Jaccard index = 0.13) are highly associated with obesity at the pathway level. Moreover, we also identified the association at the pathway level based on the GWAS and WGS data. And, there are 6 significant pathways shared by obesity and schizophrenia, and 4 significant pathways shared by obesity and MDD. Hypergeometric and Jaccard index depict that how schizophrenia (p-value = 0.002 and Jaccard index = 0.042) and MDD gene expression (p-value = 0.004 and Jaccard index = 0.043) are correlated with obesity at the pathway level. Pathway level analysis using shared pathways suggests that the two psychiatric disorders show significant patterns of associations with obesity, suggesting possible shared pathogenic

**Table 1. The hypergeometric test is employed to determine the gene's genentic importance between two diseases.** The test was carried out on transcriptomic and GWAS/WGS data in order to validate the association between obesity and psychiatric disease at the gene levels.

| Data Type | Disease A | Disease B | Raw Genes of Disease A | Significant Genes of Disease B | Raw Genes of Disease B | Significant Genes of Disease B | Total Raw Genes | Shared Significant Genes | Hyper-geometric Value |
|---|---|---|---|---|---|---|---|---|---|
| Transcriptomic | Obesity | Schizophrenia | 30315 | 1517 | 25213 | 2368 | 55528 | 162 | 1.01e-26 |
| Transcriptomic | Obesity | Depression | 30315 | 1517 | 19150 | 2219 | 49465 | 246 | 1.50e-70 |
| GWAS/WGS | Obesity | Schizophrenia | 1366 | 1130 | 1961 | 1836 | 3202 | 126 | 1.15e-568 |
| GWAS/WGS | Obesity | Depression | 1366 | 1130 | 446 | 406 | 1812 | 43 | 3.24e-244 |

**Table 2. A hypergeometric test and Jaccard index was used to determine the genentic relevance of the signaling pathway between two diseases.** The test were run on transcriptomic and GWAS/WGS data to confirm the association between obesity and psychiatric disorders at the pathway level.

| Data Type | Disease A | Disease B | Pathways of Disease A | Significant Pathways of Disease B | Pathways of Disease B | Significant Pathways of Disease B | Total Pathways | Shared Significant Pathways | Hyper-geometric Value | Jaccard index |
|---|---|---|---|---|---|---|---|---|---|---|
| Transcriptomic | Obesity | Schizophrenia | 1930 | 278 | 2246 | 301 | 4176 | 134 | 1.45e-88 | 0.3 |
| Transcriptomic | Obesity | Depression | 1930 | 278 | 2137 | 287 | 4067 | 65 | 1.11e-19 | 0.13 |
| GWAS/WGS | Obesity | Schizophrenia | 1488 | 50 | 2108 | 96 | 3596 | 6 | 0.0019 | 0.042 |
| GWAS/WGS | Obesity | Depression | 1488 | 50 | 943 | 46 | 2431 | 4 | 0.003512 | 0.043 |

mechanisms for transcriptomic data. However, Pathwasy level relation for GWAS/WGS analysis are comparatively lower than transcriptomic analysis. Therefore, patient specific transcriptomic data outperformed over SNPs based GWAS/WGS data.

## Cluster analysis of the shared significant genes (Co-expression) between obesity and various psychiatric disorders

Co-expression and cluster analysis are performed on the transcriptomic count data of obesity. 179 shared singnificant genes are considered for the analysis that are common between obesity and both psychiatric disorder from transcriptomic, WGS, and GWAS data. We then performed co-expression analysis, in which we considered transcriptome count values for the selected genes. Fig 7A illustrates the correlation matrix for the shared genes using pearson correlation method. The correlation matrix assign a value ranges from -1 to 1. However, positive correlation depicts the biological importance of co-expressed genes [77]. Then, based on the co-expression, hierarchical clustering was performed to group the genes that were expressed in the same direction. Thus, we have found 4 significant clusters that are strongly expressed in transcriptomic raw data as sown in Fig 7. In these 4 regions of genes, the correlation value was > 0.7. Fig 7B–7E contain 25, 21, 17 and 18 genes respectively. The co-expression analysis along with hierarchical features allows to define the biological mechanism of the genes [78, 79]. Then, we have matched the shared differential transcriptomic results with the co-expression outcomes of transcriptomic raw data. The co-expression and cluster analysis of transcriptome raw data validate the differential transcriptomic analysis of our study [80]. The highly expressed gene by which obesity may interact with and influence the development of schizophrenia/MDD and vice versa.

## Validation of differential transcriptomic analysis by GWAS/WGS and co-expression cluster analysis

Transcriptomic analysis found that obesity patient shared 163 and 247 biomarker genes with schizophrenia and major depressive disorder. GWAS and WGS data provided 126 shared genes between obesity and schizophrenia; and 43 genes between obesity and mdd. In order to validate the shared transcriptomic profile of a pair (such as scz and obesity), we have compared with the same pair's shared biomarkers resulted in GWAS and WGS data. Thus, 163 shared biomarker genes of obesity and scz from transcriptomic analysis are compared with the 126 shared biomarkers of the same pair in GWAS/WGS analysis. As a result, there are three genes (ETS1, RGS6, BRINP2) that are found in both analyses for the pair. Similarly, one common gene (AK8) found between obesity and mdd pair in both analyses.

Afterward, we have compared the transcriptome shared significant signaling pathways and GO terms to the shared significant signaling pathways and GO terms identified in the GWAS/

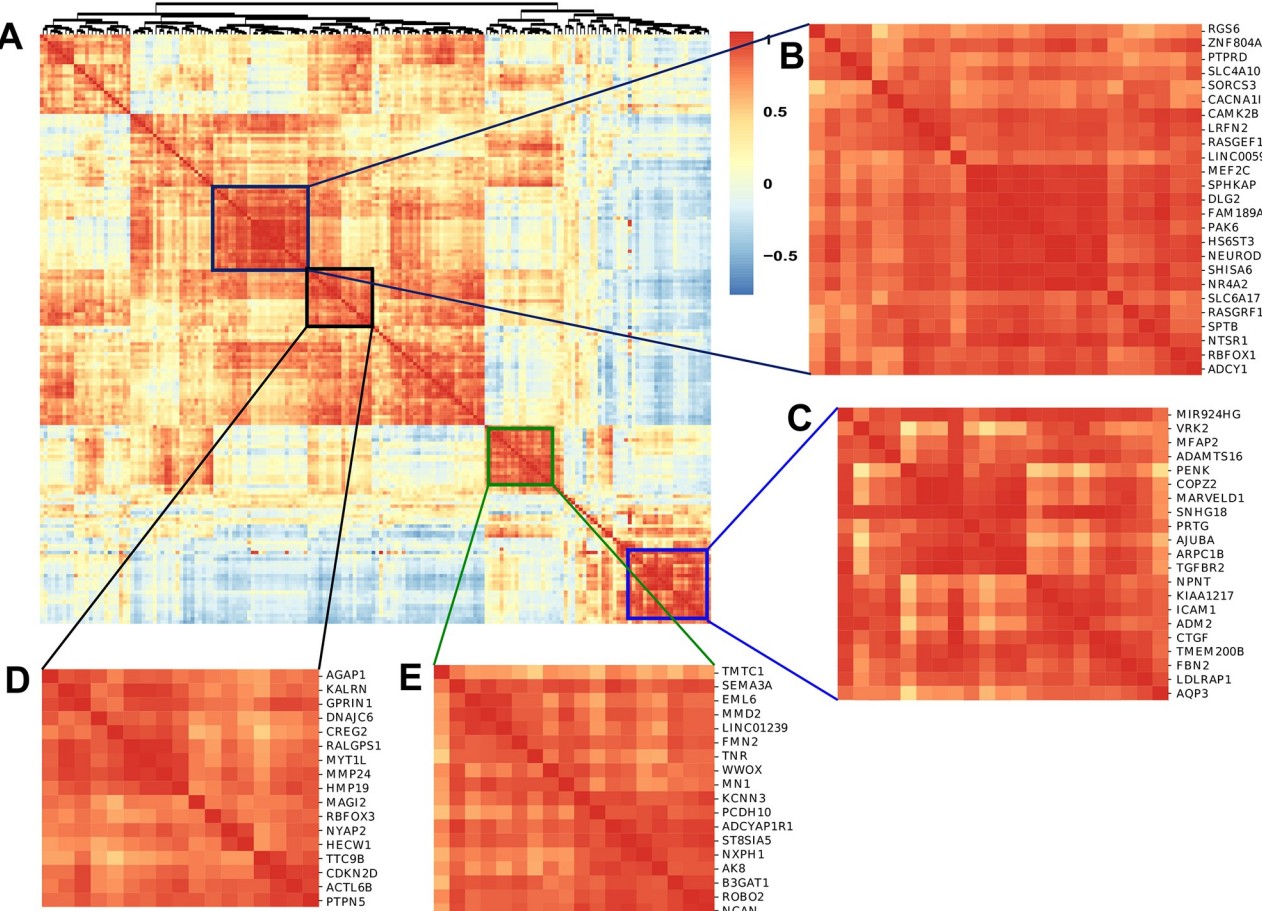

**Fig 7.** A. cluster analysis of the shared co-expressed genes between obesity and psychiatric disorders in all the selected transcriptomic data. In our study, four clusters were selected using co-expression value > 0.7. B. Cluster one contains 25 genes C. The second cluster contains 21 genes D. The third cluster contains 17 genes E. The fourth cluster contains 18 genes co-expressed together respectively.

WGS data pairwise. There are 151 shared significant pathways of obesity and mdd in transcriptomic analysis and that are compared with the 243 shared significant pathways in GWAS/WGS analysis. Whereas we have found 10 common signalling pathways and 2 significant GO term between obesity and depression for both analysis. As well as obesity shared 3 common significant signalling pathways, but no significant GO terms with schizophrenia in this comparison. The shared significant signaling pathways that are commonly found between obesity and schzophrenia in both analyses are: Neuronal System (R-HSA-112316), Platelet homeostasis (R-HSA-418346), Phosphodiesterases in neuronal function WP4222. Similarly, the shared signaling pathways between obesity and major depressive disorder in both analyses are: Adrenergic signaling in cardiomyocytes, Arrhythmogenic right ventricular cardiomyopathy, Transmission across Chemical Synapses (R-HSA-112315), Neuronal System (R-HSA-112316), Axon guidance (R-HSA-422475), Developmental Biology (R-HSA-1266738), Cardiac conduction (R-HSA-5576891), Phase 0—rapid depolarisation (R-HSA-5576892), SALM protein interactions at the synapse (R-HSA-8849932), Signaling by ERBB4 (R-HSA-1236394). The shared Gene Ontology terminology between obesity and depression in both analyses are: protein localization to cell periphery (GO:1990778), protein localization to plasma membrane (GO:0072659).

Moreover, we have combined the transcriptomic and GWAS/WGS analysis; and found 179 biomarker genes are shared among the diseases. Then, the co-expression analysis of the transcriptomic data are evaluated with the outcomes of differential expressed analysis. In that case, 81 genes are highly expressed in co-expression cluster analysis (with co-expression value > 0.7) among 179 significant DEGs (as shown in Fig 7).

## Discussion

In our study, we have developed a pipeline considering various statistical and bioinformatics software as shown in Fig 1. First of all we have performed transcriptomic analysis of the selected diseases datasets. Then, we have identified shared significant DEGs (Fig 2) from transcriptomic analysis which used to determine significant signalling pathways, GO (Figs 5 and 6) and gene-disease associations network (Fig 4). Thus, this findings help us to find the genes and molecular mechanisms that are common in obesity and its associated psychaitric disorder such as schizophrenia and major depression disorder. Obese patients are more likely than non-obese people to develop a number of psychiatric disorders [81], but whether there are pathophysiology in obese patients that affect psychiatric disorders is still unclear. Furthermore, schizophrenia patients who use 2nd-generation antipsychotics are more likely to gain weight because they consume more calories [82]. Similarly, major depression has bidirectional relation with obsity [40]. If there are pathways that shared by these conditions, then this findings will tell us a great deal about these relations, and provide new therapeutic interventions.

In this study, we have examined transcriptomic data derived from neuronal cells of obese individuals in contrast to non-obese. These data were derived from both post mortem brain specimens and hypothalamic neurons generated from iPS cells [59]. The use of iPS cells allows the examination of neuronal cells from healthy individuals where tissue samples from the brain cannot be obtained; the iPS cells are then stimulated to form neuronal cells. However, it should be noted that the nature of these will have important differences to that of neuronal tissues sampled from the brain [83]. For iPSC-derived neurons, which (unlike samples from the brain) are generated in vitro [84], however, differences between iPS cells derived from obese and non-obese patients will reflect the intrinsic genetically defined difference between them. Similar iPS-derived neuron transcriptome data were generated from patients with schizophrenia and with MDD, and their respective controls [63, 64]. This data allows us to look for transcriptomic patterns that the obese and psychiatric patients have in common.

As far as we know, this research is the first study which conduct the transcriptomic and genomic comparison of obese and psychiatric disorders. The Fig 5 shows the top 25 signaling pathways and GO terms between obesity and schizophrenia that were found by transcriptomic analysis. A substantial association has been found between NrCAM and schizophrenia, and it's a gene among 12 CAM genes that involved in the CAM pathway [86]. In our study, Neuronal Cell adhesion molecule (NrCAM interactions) are highly expressed in both scz and obesity patient's central nervous system(with a p-value of 0.00028). NrCAM's functional alterations, and disturbance of interactions with other proteins cause neurodevelopment disorders such as schizophrenia [85, 86]. CAM pathway's present is also identical in obesity patient [87] including NrCAM gene [88, 89].

Neurotransmitter release (GO:0098700) is accomplished by endocytosis recycling of synaptic vesicles (GO:0036465). And the fictionalisation of brain are controlled by neurotransmitter, however, neurotransmitter level changes cause neurodevelopment disorders (scz, mdd, bipolar) [90, 91]. Our research discovered that the release cycle of neurotransmitters such as serotonin and dopamine is associated with the neural system of scz patients [63]. Similarly, in obese

individuals, the release of pituitary hormone was activated by the serotonin-neurotransmission cycle [92]. Moreover, Dopamine and Serotonin is responsible for controlling the hunger in both men and women [93–95]. Fig 6 represents the top 25 signalling pathways and GO terminology from transcriptomic analysis between obesity and mdd. The Neuronal System (R-HSA-112316) is a high-level pathway that governs the nervous system's processes. This pathway is significant across all the datasets (transcriptome and GWAS/WGS) and neuropsychiatric disorders in our study [96–98]. In the same case, the neural system regulates one's eating patterns [99]. In differential transcriptomic analysis and GWAS/WGS analysis, the neuronal system's low level signalling pathways called "Transmission across chemical synapses—(R-HSA-112315)" highly enriched among obesity, schizophrenia and depression patients [100–102]., the GO term chemical synaptic transmission (GO:0007268) is substantially enriched, with a p-value of 8.66E-09 [98]. Neurotransmitter Release Cycle (R-HSA-112310) is also detected in mdd and obesity (together with scz and obesity) [103], even through significant enrichment of Dopamine Release Cycle (R-HSA-212676) with a p-value of 0.00027 [104, 105]. Ca2+ flow across the NMDA receptor activates the renin-angiotensin system (RAS) which increased obesity condition [106]. "Ras activation uopn Ca2+ infux through NMDA receptor—(R-HSA-442982)" pathway is significantly associated between obesity and mdd, according to our findings. Recently, the RAS system has been identified as the most likely blocking candidate for depression and anxiety reduction [107, 108]. Overconsumption or increasing glutamate level result in hunger, which eventually leads to obeisy [109, 110]. Similarly, the amount of glutamate in mdd patients is greater than in healthy persons. Our study found glutamate binding and activation (R-HSA-438066, R-HSA-442755 and R-HSA-438064) [111–113]. These findings are consistent with our understanding from clinical and biochemical research that obese individuals have higher levels of neurotransmitter alterations and increased neurotransmitter Stimulator such as glutamate; as well as the findings suggest that this is also true for schizophrenia and MDD patients. This immediately suggests the possibility that NMDA-receptor antagonist (ketamine) and co-agonist (d-serine) may be a useful antidepressant therapeutic approach to reduce risks of scz and mdd patients [114–116], although it should be noted that the cause and effect need to be further examined. Moreover, d-serine reduce the appetite for food in obese people [117]. Indeed, further clinical verification will be required to confirm that this are considerable therapeutic approaches.

We identified a diseasome network (Fig 4) and Venn diagram (Fig 2A) of transcriptomic data to identify the shared genes as well as uncover the pathways that are responsible for the diseases. Differential analysis of transcriptomic data found 51 common significant genes among all the disease (obe, scz and mdd); and 13 shared significant genes are found from GWAS and WGS data (Fig 3). In case of GWAS data, the number of significant genes for MDD is comparatively fewer than that of obesity and schizophrenia, as seen in Fig 3. As a result, MDD may not reveals much genomic functional diversity compared to schizophrenia and obesity based on GWAS/WGS data. Therefore, the correlation between obesity and schizophrenia based on GWAS data is much better, counting the number of shared genes between two diseases. Although GWAS data reported (Fig 3) that the correlation between obesity and schizophrenia is significant, but transcriptomic data reveals (Fig 2A) that the the number of shared genes between obesity and MDD are comparatively higher. GO and signalling pathways-based analysis also support the comparison of transcriptomic and GWAS/WGS analysis considering the number of shared significant signalling pathways and GO terminologies between obesity and either of the psychiatric disorders. It follows that a higher level of correlation exists between obesity and psychiatric disorder (scz and mdd) based on the transcriptomic differential expression analysis; and its corresponding GO, and signalling pathways.

Then the hypothesis is validated by comparing differential expression analysis of transcriptomic data with GWAS/WGS data as well as co-expression network analysis. Comparison of differential transcriptomic profiles and GWAS/WGS profiles identified 3 significant shared genes (ETS1, RGS6, BRINP2) commonly present between obesity and schizophrenia pair in both analyses. Previous study also proved that these 3 genes are highly expressed in both obesity and scz. ETS1 is highly expressed in obese case compared to healthy persons and its a potential therapeutic of obese patients [118–120]. Similarly, both transcriptome and GWAS data have demonstrated the ETS1 expression alterations in schizophrenia patients [121–123]. Transcriptomic factor 'RGS6' are in charge of unusual food consumption [124] and abdominal obesity in response to psychological and social strain [125]. One of the most important molecules in the central nervous system is 'RGS6', which is found in a lot of psychiatric disorders like depression and schizophrenia [126, 127]. Likewise, it has been discovered that BRINP2 is overexpressed and shared both in obesity and schizophrenia [128, 129]. But we have got only one shared gene (AK8) by comparing differential transcriptomic analysis and GWAS/WGS data for the obesity and depression pair. Similarly, shared significant pathways and GO terms (as described in sec) validate our findings and demonstrate important functional genomics and molecular mechanism for obestiy and psychiatric disorder. Although combined transcriptomic and GWAS/WGS profiles could extract more information regarding the common molecular mechanisms between obesity and psychiatric disorders, still transcriptomic related data are more reliable as patient-specific iPSC data collected from living organ/cells. Furthermore, GWAS's may not provide proper etiological and biological meaning of a disease as the data is non-coded and pleiotropic [130]. However, we have considered all the transcriptomic and GWAS/WGS genes that are significantly expressed and shared among all the diseases to perform co-expression cluster analysis (Fig 7). Co-expression cluster analysis found 81 genes are highly expressed. The co-expression network analysis successfully validate our findings from transcriptomic and GWAS/WGS data. As well as these clustered genes will elucidate more details about the complex common etiology, functional genomics and molecular mechanism between obesity and psychiatric disorder in future.

To substantiate our hypothesis regarding the identified signaling pathways, we relied on a few previous studies. It implies that obesity has an impact on the neural system that are also common in psychiatric disorders; whereas it communicates with other neurons through chemical synapses by neurotransmitter release such as dopamine, seratonin and glutamate [131, 132]. According to the pathway and GO study, the impact of stress in functional genomics alterations in schizophrenia and MDD is almost identical to the impact of functional genomics alterations in obesity [35]. Alterations to the common functional genomics associated with obesity and the two psychiatric disorders result in an excessive amount of pro-inflammatory cytokines circulating in nervous system that produce and changes neurotransmitter level. This unregulated inflammation in the immune system may trigger unusual behaviour associated with psychiatric disorder [133]. Therefore, our identified common genetic variants and pathways-based analysis represents a novel approach to develop therapeutics that can lower the risk of obesity, depression and schizophrenia rivera2015role.

It should be noted that the major limitations of this study is the availability of samples from living brain cells as well as relatively very few data samples for each disease. Moreover, we didn't consider sex, age, ethnicity and other features of the samples in our study. As a result, further validation is essential to fully assess the biological relevance of the identified possible target candidates in this research.

## Conclusion

In this study, an integrated bioinformatics approach was designed for identifying potential risk factors of psychiatric disorders like schizophrenia and MDD associated with obesity. RNA-seq data was collected from brain-sampled neurons (post mortem), iPS cell-derived neurons, and GWAS/WGS collected from open-access databases (NCBI). We have identified that the iPSC-derived transcriptomic profile of obese patients has an effect on the neuronal system, changes in neurotransmitter levels and release, such as dopamine, glutamate, and serotonin. This could lead to schizophrenia or depression in obese patients, or the other way around in different situations. Furthermore, we have conducted GWAS/WGS and co-expression cluster analysis to validate and compare our differential expression analysis of transcriptome data. However, differential transcriptome analysis consistently outperformed other analyses at the gene or pathway level. This implies the association of obesity with psychiatric disorders that may reflect changes and disturbances in genomics and molecular pathology. The above study could help to develop therapeutic approaches to ameliorate the condition of obesity-affected psychiatric patients.

## Author Contributions

**Conceptualization:** Md Khairul Islam, Md Habibur Rahman, Md Zahidul Islam, Mohammad Ali Moni.

**Data curation:** Md Khairul Islam, Md Rakibul Islam, Md Habibur Rahman, Mohammad Ali Moni.

**Formal analysis:** Md Khairul Islam, Md Rakibul Islam, Md Habibur Rahman, Md Zahidul Islam, Mohammad Ali Moni.

**Investigation:** Md Khairul Islam, Md Zahidul Islam, Md Mehedi Hasan, Md Mainul Islam Mamun, Mohammad Ali Moni.

**Methodology:** Md Khairul Islam, Md Habibur Rahman, Mohammad Ali Moni.

**Project administration:** Md Zahidul Islam, Mohammad Ali Moni.

**Resources:** Mohammad Ali Moni.

**Software:** Md Khairul Islam.

**Supervision:** Md Khairul Islam, Md Habibur Rahman, Md Zahidul Islam, Mohammad Ali Moni.

**Validation:** Md Rakibul Islam, Md Habibur Rahman, Md Zahidul Islam, Md Mehedi Hasan, Mohammad Ali Moni.

**Visualization:** Md Khairul Islam.

**Writing – original draft:** Md Khairul Islam.

**Writing – review & editing:** Md Khairul Islam, Md Rakibul Islam, Md Habibur Rahman, Md Zahidul Islam, Md Mehedi Hasan, Md Mainul Islam Mamun, Mohammad Ali Moni.

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
