## [Decision Letter · Decision Letter 0]

15 Mar 2022

PONE-D-21-36929Integrated Bioinformatics Approaches to Identify the Pathobiological Associations of Schizophrenia and Major Depressive Disorder with ObesityPLOS ONE

Dear Dr. Moni,

Thank you for submitting your manuscript to PLOS ONE. After careful consideration, we feel that it has merit but does not fully meet PLOS ONE’s publication criteria as it currently stands. Therefore, we invite you to submit a revised version of the manuscript that addresses the points raised during the review process.

We look forward to receiving your revised manuscript.

Kind regards,

Zezhi Li, Ph.D., M.D.

Academic Editor

PLOS ONE

Journal Requirements:

2. Please update your submission to use the PLOS LaTeX template. The template and more information on our requirements for LaTeX submissions can be found at http://journals.plos.org/plosone/s/latex

3. Thank you for including your ethics statement: "Obesity patient iPS cell RNA-seq transcriptomic datasets were examined in this study. The dataset was generated from induced pluripotent stem cells of hypothalamic/motor neurons of obese patients and completed by the Board of Governors Regenerative Medicine Institute in Los Angeles, California [41]. The dataset’s GEO accession number is GSE95243, and all the curated datasets are in the public domain, namely the NCBI database (https://www.ncbi.nlm.nih.gov/). The study includes 7 cell samples derived from hypothalamic neurons from healthy individuals and 2 cell samples derived from motor neurons. Case study 5 samples were collected from obese patients and 5 samples were obtained from post-mortem, both derived from hypothalamic neurons. RNA-seq transcriptomic data extracted from the samples (using QIAGEN RNeasy mini kits), sequenced, and compared between control vs case samples to obtain insights into the transcriptional dysfunction in the sampled cells from diseased individuals. Aside from obesity-related data, we have analyzed RNA-Seq transcriptomic data from schizophrenic individuals (and normal controls) from NCBI GEO accession number GSE92874 [42]; SUNY Pathology and Anatomical Sciences Department researchers produced the data. The study included 4 healthy patient samples and 4 schizophrenia patient samples; the samples were generated iPS cells. Another study similarly employed RNAseq data from iPS cells from MDD and control individuals collected from NCBI GEO accession number GSE125664 [43]. This study was conducted by Max Planck Institute of Immunobiology and Epigenetics, Freiburg im Breisgau, Germany. Six major depressive disorder patients and three healthy patient samples were collected from iPSC-derived neurons.

The datasets were processed to identify genes diﬀerentially expressed between individuals aﬀected by these conditions and their respective unaﬀected controls. RNA-seq transcriptomic data was processed by the DESeq2 package using a negative binomial distribution assumption, and the diﬀerential expression was determined from microarray data processed by the limma package. We then performed quantile normalization to eliminate platform technology-related variation and data noise [44]. The diﬀerentially expressed genes were then ﬁltered using two conditions to identify the most important genes. The ﬁrst was a p-value threshold less than 0.05, and the second was an absolute log2 fold change greater than 1 (— log2fc — ≤ 1)".   

(1) For studies reporting research involving human participants, PLOS ONE requires authors to confirm that this specific study was reviewed and approved by an institutional review board (ethics committee) before the study began. Please provide the specific name of the ethics committee/IRB that approved your study, or explain why you did not seek approval in this case.

(2) Please provide additional details regarding participant consent. In the ethics statement in the Methods and online submission information, please ensure that you have specified (1) whether consent was informed and (2) what type you obtained (for instance, written or verbal, and if verbal, how it was documented and witnessed). If your study included minors, state whether you obtained consent from parents or guardians. If the need for consent was waived by the ethics committee, please include this information.

Reviewers' comments:

Reviewer's Responses to Questions

**Comments to the Author**

1. Is the manuscript technically sound, and do the data support the conclusions?

Reviewer #1: Yes

Reviewer #2: Partly

2. Has the statistical analysis been performed appropriately and rigorously? 

Reviewer #1: Yes

Reviewer #2: Yes

3. Have the authors made all data underlying the findings in their manuscript fully available?

Reviewer #1: No

Reviewer #2: Yes

4. Is the manuscript presented in an intelligible fashion and written in standard English?

Reviewer #1: Yes

Reviewer #2: No

5. Review Comments to the Author

Reviewer #1: The manuscript entitled " Integrated Bioinformatics Approaches to Identify the Pathobiological Associations of Schizophrenia and Major Depressive Disorder with Obesity" investigates the genetic interaction of Obesity with Schizophrenia and Major Depressive Disorder. Given that people with Obesity are more likely to have Schizophrenia and Major Depressive Disorder, such studies can help to determine therapeutic targets to minimize the likelihood of the occurrence of the target diseases. Some gained results are very intriguing and significant but still, some concerns remain:

01. There are some inconsistencies like typo errors in the writing of the manuscript. Please address them.

02. There are several psychiatric disorders but Why the authors chose these two particular psychiatric disorders is not clear. The authors should explain more details in the introduction sections.

03. Although the authors have provided the rationale behind considering datasets from iPSCs samples. More description is required on the datasets in the materials and Methods sections.

04. The authors should provide relevant recent references in the introduction and discussion sections.

05. Discussion section needs to improve and need to compare with the recent similar work.

06. Which disease among the two psychiatric disorders is mostly affected by obesity? Need to discuss in the discussion section.

06. The reason behind clustering co-expressed genes should be more clear.

07. How sensitive is the method to a low sample size such as schizophrenia and major depressive disorder?

08. Line 33-39 is not clear

09. What's the reason behind taking the 4 highlighted cluster areas only?

10. Why threshold-based cluster analysis rather than k-means is used, the author should explain more details.

Reviewer #2: 1) The results are not sufficient to support the conclusion. The author needs to make more analysis and interpretation of these results.

2) The manuscript needs to revise the language to make readability.

More details please see the reviewer attachment.

6. PLOS authors have the option to publish the peer review history of their article (what does this mean?). If published, this will include your full peer review and any attached files.

Reviewer #1: No

Reviewer #2: No

---

## [Author Response · Author response to Decision Letter 0]

7 Jun 2022

Response to Reviewers

Reviewer 01

The manuscript entitled " Integrated Bioinformatics Approaches to Identify the Pathobiological Associations of Schizophrenia and Major Depressive Disorder with Obesity" investigates the genetic interaction of Obesity with Schizophrenia and Major Depressive Disorder. Given that people with Obesity are more likely to have Schizophrenia and Major Depressive Disorder, such studies can help to determine therapeutic targets to minimize the likelihood of the occurrence of the target diseases. Some gained results are very intriguing and significant but still, some concerns remain:

01. There are some inconsistencies like typo errors in the writing of the manuscript. Please address them.

Answer: Thank you for your comments. We have revised our manuscript and updated it according to your suggestions which can be found in the revised version.

02. There are several psychiatric disorders but Why the authors chose these two particular psychiatric disorders is not clear. The authors should explain more details in the introduction sections.

Answer: Thank you for your comments. We have modified our manuscript according to your comment which is shown as the highlight. 

03. Although the authors have provided the rationale behind considering datasets from iPSCs samples. More description is required on the datasets in the materials and Methods sections.

Answer: Thank you for your comments. We have revised our manuscript and added the required description in the introduction and discussion section for your kind consideration.

04. The authors should provide relevant recent references in the introduction and discussion sections.

Answer: Thank you for your comments. We have modified our manuscript according to your comment which is shown as the highlight. 

05. Discussion section needs to improve and need to compare with the recent similar work.

Answer: Thank you for your comments. We have revised and modified our manuscript according to your comment which is shown as the highlight. 

06. Which disease among the two psychiatric disorders is mostly affected by obesity? Need to discuss in the discussion section.

Answer: Thank you for your comments. We have revised and modified our manuscript according to your comment which is shown as the highlight.

06. The reason behind clustering co-expressed genes should be more clear.

Answer: Thank you for your comments. We have revised our manuscript and added more descriptions regarding co-expression cluster analysis both in the method and result section.

07. How sensitive is the method to a low sample size such as schizophrenia and major depressive disorder?

Answer: Thank you for your comments. The sample size should be at least 3 to get the minimum computational power of the transcriptome differential analysis. In our study, our selected dataset meets the minimum requirements. However, it is still a drawback for your study with limited sample availability and specified the fact in the discussion section.

08. Line 33-39 is not clear

Answer: Thank you for your comments. We have revised and modified our manuscript according to your comment which is shown as the highlight.

09. What's the reason behind taking the 4 highlighted cluster areas only?

Answer: Thank you for your comments. The highlighted regions are highly expressed with a co-expression value greater than 0.7. Therefore we have considered only these four regions to validate the differential expression analysis. 

10. Why threshold-based cluster analysis rather than k-means is used, the author should explain more details.

Answer: Thank you for your comments. K-means needs to specify the number of clusters initially. Moreover, it performed well on large datasets, but we have considered only the significant genes found in transcriptomic, GWAS, and WGS data. Hierarchical clustering analyses are especially useful when the target is to arrange the clusters into a natural hierarchy.

Reviewer 02

“Integrated Bioinformatics Approaches to Identify the Pathobiological Associations of Schizophrenia and Major Depressive Disorder with Obesity”. The author applied public datasets of transcriptome and GWAS/WGS to investigate the shared genes and pathways between obesity and SZ, as well as obesity and MDD. The idea of the study is interesting; however, the results are not strong enough to support their conclusion. 

Thank you for your comments

Major issues:

 1. Obesity could not be considered as a major risk of SZ. The obesity existed in SZ patients is more likely due to the use of psychotropic medications such as second-generation antipsychotics. Hence, the transcriptome and genetic association study between obesity and SZ should be take a careful consideration.

Answer: Thank you for your comments. We have carefully modified our manuscript considering the fact you mentioned. Major depression and obesity have bidirectional relation, whereas schizophrenia patient has the possibility of being diagnosed with obesity due to antipsychotic medications. However, maternal obesity has been associated with schizophrenia in females. As a result, we've demonstrated every possible association between obesity and psychiatric disorders.

 2. The author involved iPSC from obesity subjects, and these iPSC were differentiated into hypothalamic neurons. Hypothalamic is a central coordinator of obesity. However, the iPSC used for SZ and MDD were finally differentiated into forebrain neurons. Actually, the authors compared the differently expressed genes between different neurons. Although they found some overlapped genes, especially those changed in the same direction, those genes could be functioned in the basic biological process of neurons. In fact, the GO enrichment and pathway results in this study also showed this problem---they are mostly enriched in synaptic associated processes. Hence, such comparison could not provide comprehensive evidence about the common biological mechanism shared between obesity and psychiatric disorders.

 3. 

Answer: Thank you for your comments. Obesity and psychiatric has a relation and it is evident by common molecular function found in patients. Proper datasets' unavailability is one of the major drawbacks of our study. However, we have carefully collected the data that can provide a deep understanding of the common molecular mechanism between obesity and psychiatric disorders. In the introduction section, we have demonstrated the reasons behind selecting hypothalamus data with proper references and compared it with forebrain neurons of SCZ and MDD data. We also included references to prove the function of synaptic related processes in obesity and psychiatric disorders both in the introduction and discussion section.

 4. The GWAS and WGS data might provide more convincing association between obesity and psychiatric disorders, however, there are not many results and interpretation about these datasets. In addition, the analysis method of this part is not clear and missing important description. For example, when combing the large dataset for analysis, how the author handled with the confounding factors, such as ethnicity, age, sex, drug dosage, smoking and others?

Answer: Thank you for your comments. We have added more information relating to GWAS and WGS. We have also discussed the drawback of GWAS/WGS data. In our study, we have used the GWAS/WGS data to validate the findings as well as the limitation of GWAS/WGS. One of the major drawbacks of our study is that we didn’t consider the fact of ethnicity, age, sex, etc for either transcriptomic or GWAS/WGS data. We have modified our manuscript accordingly and highlighted it for your kind consideration.

Other issues:

Overall, the author needs to revise the language to improve readability.

Answer: Thank you for your comments. We have revised the manuscript accordingly.

Title: the title is confusing. It seems to study the samples of schizophrenia with obesity and samples of MDD with obesity.

Answer: Thank you for your comments. We have compared the relationship between obesity and psychiatric disorders such as schizophrenia and depression. However, we have modified the title to depict our hypothesis more clearly. “Integrated Bioinformatics and Statistical Approach to Identify the Common Molecular Mechanisms of Obesity that are linked to the development of two Psychiatric Disorders: Schizophrenia and Major Depressive Disorder.” 

Keywords: too much keywords. “comorbidities” is not relevant to this study, since subjects the author involved in are not patients comorbidity with any other disease. 

Answer: Thank you for your observation. We have modified our manuscript accordingly.

Abstract:

 1. “Obesity is a major risk factor for both schizophrenia and MDD.” The author need pay attention to the phrase of “a major risk factor”, especially for schizophrenia. There is no evidence indicate that obesity is a major risk factor for SZ. The author need double check this statement.

Answer: Thank you for your comments. Considering the fact of SCZ, we have changed the sentence, “Epidemiologic data indicate that Obesity is acting as a risk factor for a neuropsychiatric disorder such as schizophrenia, major depression disorder, and vice versa.”

 2. “… in patients with psychiatric disorders that are also relevant to the pathogenicity of obesity.” The author need correct this sentence. It makes readers to think these are patients comorbidity with obesity. 

Answer: Thanks for your comments. We have modified our manuscript according to your comment which is shown as the highlight. “To address this issue, we have developed a pipeline that integrates bioinformatics and statistical approaches such as transcriptomic analysis to identify differentially expressed genes (DEGs) and molecular mechanisms in patients with psychiatric disorders that are also common in obese patients.”

 3. The last sentence “we also validated our findings by use of GWAS and WGS”. It is confused how the author validated and confirmed their findings? Since there is no any result showed the overlap between the DEGs found in transcriptomic analysis and those found in GWAS/WGS. “Confirmed the likely involvement” this statement is also ambiguous. 

Answer: Thank you for your comments. Unfortunately, we have missed the part, so we have added section 3.7. Also, modified the sentence accordingly. “Finally, we also validated our findings using genome-wide association study (GWAS) and whole-genome sequence (WGS) data from SCZ, MDD, and OBE. We confirmed the likely involvement of four significant genes both in transcriptomic and GWAS/WGS data. Moreover, we have performed co-expression cluster analysis of the transcriptomic data and compared it with the results of transcriptomic differential expression analysis and GWAS/WGS.”

Introduction:

1st p: Revise the paragraph to make it brief and clear for reader to understand obesity and the aim of study. Too much irrelevant information inside.

Answer: Thank you for your comments. We have modified our manuscript according to your comment which is shown as the highlight. 

2nd p: ref 17 and 18: no any information about the association between obesity and schizophrenia indicated. The author should correct the refs. 

Answer: Thank you for your observation. We have modified our manuscript according to your comments which is shown as the highlight. 

3rd-4th p: The author needs involve more information about their association with obesity, not just explain the two diseases. Importantly, the author needs to explain why they choose two psychiatric disorders to analysis? What is their relevance? 

Answer: Thank you for your comments. We have modified our manuscript according to your comment which is shown as the highlight. 

5th p, line 76: Please check the ref 17. It could not find any information about the Chinese ethnicity in this ref.

Answer: Thank you for your comments. We have corrected the ref.

6th-7th p: It is confused why the author talked much about endocrine systems with obesity and psychiatric disorders. How the endocrine systems relate to the study aim? In addition, line 91-97, this long sentence is very hard to read. Line 97-98: this sentence is also hard to read. How “these 70 candidate genes” comes from?

Answer: Thank you for your comments. The endocrine system is described because obesity is frequently accompanied by neuroendocrine changes which trigger for mood episodes, psychosis exacerbation, and cognitive decline. We also modified the lines according to your comments.

8th p: line 107-109: it is very hard to follow the logic of this sentence. 

Answer: Thank you for your comments. We have simplified the sentences for the reader.

9th p: line 122: missing WGS data. In addition, the author should explain in the introduction what kind of data they were used in this study and why such iPSC data were selected for this study. 

Answer: Thank you for your comments. We have modified our manuscript according to your comment which is shown as the highlight. 

Method: 

 5. Page 7, line 166: the author need make some comments why they did not do the FDR correction for DEG analysis. 

Answer: Thank you for your comments. We have compared the transcriptomic DEGs data with GWAS/WGS data. Most of the GWAS/WGS databases provide processed data and they don’t include FDR correction value. P-value was considered to compare two variables in a comparable pattern.

 6. Page 7, line 168-177: Please make a more clear and detailed description about how this disease-gene network was conducted. It is hard to follow. 

Answer: Thank you for your comments. We have modified our manuscript according to your comment which is shown as the highlight. 

 7. Page 7, section 2.2: Please make a more detailed description about the sample for GWAS and WGS. Such as how many datasets/samples are used, what are their biological source. What kind of mutation information was involved for analysis? In addition, line 197-199: it could not find any information about the source of WGS data. Importantly, when combing the large dataset for analysis, how the author handled with the confounding factors, such as ethnicity, age, sex, drug dosage, smoking and others?

Answer: Thank you for your comments. We have collected the data from various sources such as the GWAS catalog, PheGenI, dbGaP, UK-Biobank, and Clinver (from NCBI). The corresponding articles for each database include all details regarding the samples and biological sources. We didn’t include the details to make it more clear and more specific. Moreover, we ignore the confounding factors both for transcriptomic and GWAS/WGS data. We have described it as one of the limitations of our study. 

 8. Page 8, line 190: “so may influence the [46]”. Please complete the sentence. 

Answer: Thank you for your comments. We have modified our manuscript based on your comments.

 9. Page 8, line 191: it was confused why the p-value criteria was different from the result. (1.0e-6 vs 1.0e-5) 

Answer: Thank you for your comments. We have revised our manuscript in response to your comments, which are highlighted.

 10. Page 9, line 243-245: why the author also analyzed the gene data correlated between SZ and MDD? It seems not the study aim and there was also no any interpretation about the result. 

Answer: Thank you for your comments. It is true that our intention is not to identify common molecular mechanism between two psychiatric disorers. Therefore, we removed the regarding comparison part and modified our manuscript according to your comments.

Result:

 1. The author need display each figure number beside the relevant result to make readers easy follow. 

Answer: Thank you for your comments. We have modified our manuscript according to your comments.

 2. Page 10, line 27: “including that…” this sentence is hard to follow. How many genes changed in the same direction? What are their function and involved pathways? These genes could be the key genes that associated with obesity and psychiatric disorders.

Answer: Thank you for your comments. We have addressed the concerns and made changes to our manuscript in response to your comments, which are highlighted.

 3. Page 10, line 273: “obesity patient iPSCs”, the author should correct the statement. In the method, some obesity samples are from postmortem brains. In addition, line 271-275: the content of the two sentences is similar, keep one of them is better. Also, the last sentence is not clear, why the author still what to compared the genes between SZ and MDD? How this result is related to the study aim？

Answer: Thank you for your comments. We have made changes to our manuscript in response to your comments, which are highlighted.

 4. Page 10, line 278-280: the interpretation is hard to follow.

Answer: Thank you for your comments. We have modified our manuscript according to your comment which is shown as the highlight. 

 5. Figure 2B-C: It is not clear why the author split the results into two figures. The fold change and P-Value should display for every protein in the same figure. In addition, there are 52 overlapped genes in the Venn diagram. Figure B-D just involved 51 genes?

Answer: Thank you for your comments. We have visualized both p-value and fold-change in 2D using a bubble plot. Then we have represented the p-value and fold-change individually using a heatmap to more clarification of the scenario. 

Moreover, we have got 49 up-regulated and 2 down-regulated shared genes among all the diseases from our study. However, we made mistake while counting. We have revised that accordingly.

 6. Page 11, Figure 3 description: “obesity with schizophrenia and major depressive disorder.” It makes readers to think the obesity patients are comorbidity with SZ and MDD. 

Answer: Thank you for your comments. We have made changes to our manuscript in response to your comments, which are highlighted.

 7. Page 11, line 284-285: why the criteria changed, different from the method? (Page 8, line 192)

Answer: Thank you for your comments. We have made changes to our manuscript in response to your comments, which are highlighted.

 8. Page 11, line 288: what is the p-value for SZ and MDD analysis?

Answer: Thank you for your comments. We have made changes to our manuscript in response to your comments, which are highlighted.

 9. Page 11, section 3.2: how many significant genes were overlapped with the transcriptome data for obesity, SZ and MDD? 

Answer: Thank you for your comments. We have made changes to our manuscript in response to your comments, which are highlighted.

 10. Page 11, section 3.3. line 304-306: confused about this result. Why the number is different from Figure 2A. How these 44 and 80 shared genes come from?

Answer: Thank you for your comments. We have modified our manuscript and figure according to your comments and highlighted.

 11. Page 11, section 3.3. Not clear about the meaning of this part. It seems another way to explain figure 2A. It could not see how the disease -gene network works. What is the enrichment p-value of the network?

Answer: Thank you for your comments. In figure 2A we didn’t represent up and down-regulation of the shared genes. Thus, the diseasome network demonstrates the number of shared genes that changed in the same direction. 

 12. Page 12, section 3.4: The author needs to make a detailed description about this result. What GO function and pathway they received? How many shared genes were used for analysis? 

Answer: Thank you for your comments. We have added a description of the shard GO and pathways in the discussion section. And also modified section 3.4 according to your comments and the changes are highlighted for your kind consideration.

 13. Page 13, line 346-347, line 352-353: could not find the relevant figures or supplement material.

Answer: Thank you for your comments. We have added two tables corresponding to the calculations.

 14. Page 13, section 3.6: not clear how the correlation matrix generated. The author need make clear how many shared genes were used. These genes were shared among three disease or any two of them? What is the biological function of these correlated modules? In addition, why only positive correlated modules were selected? 

Answer: Thank you for your comments. We have made changes to our manuscript in response to your comments, which are highlighted both in section 2.5 and 3.6.

 15. The results from transcriptome and GWAS/WGS were displayed separately, what are their relevance. The author needs to make some comparison analysis. 

Answer: Thank you for your comments. We have added a section-3.7 to our manuscript in response to your comments, which are highlighted.

Discussion:

1st p: the author needs to make it brief and clear. Not clear why emphasis the endocrine system, how this related to the result and finding?

Answer: Thank you for your comments. We have made changes to our manuscript in response to your comments, which are highlighted.

2nd p: the author did not make a convincing explanation why the GWAS and WGS data did not perform as well as the transcriptome data. The author could discuss the impact of sample source for these datasets. 

Answer: Thank you for your comments. We have modified our manuscript according to your comment which is shown as the highlight. 

3rd p: it is confused why the author discussed much about inflammation? There were no any results showed inflammation in this study. The author should make more discussion based on their findings (GO enrichment and pathways). 

Answer: Thank you for your comments. We have made explained our findings in detail and changes to our manuscript in response to your comments, which are highlighted.

4th p: the author needs to make more discussion about the limitation of the study.

Answer: Thank you for your comments. We have made changes to our manuscript in response to your comments, which are highlighted.

---

## [Decision Letter · Decision Letter 1]

14 Oct 2022

Integrated Bioinformatics and Statistical Approach to Identify the Common Molecular Mechanisms of Obesity that are linked to the development of two Psychiatric Disorders: Schizophrenia and Major Depressive Disorder

PONE-D-21-36929R1

Dear Dr. Moni,

We’re pleased to inform you that your manuscript has been judged scientifically suitable for publication and will be formally accepted for publication once it meets all outstanding technical requirements.

Kind regards,

Zezhi Li, Ph.D., M.D.

Academic Editor

PLOS ONE

Additional Editor Comments (optional):

Reviewers' comments:

Reviewer's Responses to Questions

**Comments to the Author**

1. If the authors have adequately addressed your comments raised in a previous round of review and you feel that this manuscript is now acceptable for publication, you may indicate that here to bypass the “Comments to the Author” section, enter your conflict of interest statement in the “Confidential to Editor” section, and submit your "Accept" recommendation.

Reviewer #1: (No Response)

Reviewer #2: All comments have been addressed

2. Is the manuscript technically sound, and do the data support the conclusions?

Reviewer #1: Yes

Reviewer #2: Yes

3. Has the statistical analysis been performed appropriately and rigorously? 

Reviewer #1: Yes

Reviewer #2: Yes

4. Have the authors made all data underlying the findings in their manuscript fully available?

Reviewer #1: Yes

Reviewer #2: Yes

5. Is the manuscript presented in an intelligible fashion and written in standard English?

Reviewer #1: Yes

Reviewer #2: Yes

6. Review Comments to the Author

Reviewer #1: The current version of paper is acceptable.

Reviewer #2: The authors have adequately addressed my comments and this manuscript is now acceptable for publication. The author should correct the typos in the manuscript.

7. PLOS authors have the option to publish the peer review history of their article (what does this mean?). If published, this will include your full peer review and any attached files.

Reviewer #1: **Yes: **Dr. S.M. Hasan Mahmud

Reviewer #2: No

---

## [Editor Report · Acceptance letter]

27 Dec 2022

PONE-D-21-36929R1 

Integrated Bioinformatics and Statistical Approach to Identify the Common Molecular Mechanisms of Obesity that are linked to the development of two Psychiatric Disorders: Schizophrenia and Major Depressive Disorder 

Dear Dr. Moni:

I'm pleased to inform you that your manuscript has been deemed suitable for publication in PLOS ONE. Congratulations! Your manuscript is now with our production department. 

Kind regards, 

on behalf of

Dr. Zezhi Li 

Academic Editor

PLOS ONE